# Improvement of a 1D Population Balance Model for Twin-Screw Wet Granulation by Using Identifiability Analysis

**DOI:** 10.3390/pharmaceutics13050692

**Published:** 2021-05-11

**Authors:** Ana Alejandra Barrera Jiménez, Daan Van Hauwermeiren, Michiel Peeters, Thomas De Beer, Ingmar Nopens

**Affiliations:** 1BIOMATH—Department of Data Analysis and Mathematical Modelling, Ghent University, Coupure Links 653, 9000 Ghent, Belgium; Ingmar.Nopens@UGent.be; 2Laboratory of Pharmaceutical Process Analytical Technology, Department of Pharmaceutical Analysis, Ghent University, Ottergemsesteenweg 460, 9000 Ghent, Belgium; Michiel.Peeters@UGent.be (M.P.); Thomas.DeBeer@Ugent.be (T.D.B.)

**Keywords:** granulation, wet granulation, granules, continuous manufacturing, PAT, population balance modeling, process modeling and simulation, particle size distributions, identifiability, formulation

## Abstract

Recently, the pharmaceutical industry has undergone changes in the production of solid oral dosages from traditional inefficient and expensive batch production to continuous manufacturing. The latest advancements include increased use of continuous twin-screw wet granulation and application of advanced modeling tools such as Population Balance Models (PBMs). However, improved understanding of the physical process within the granulator and improvement of current population balance models are necessary for the continuous production process to be successful in practice. In this study, an existing compartmental one-dimensional PBM of a twin-screw granulation process was improved by altering the original aggregation kernel in the wetting zone as a result of an identifiability analysis. In addition, a strategy was successfully applied to reduce the number of model parameters to be calibrated in both the wetting zone and kneading zones. It was found that the new aggregation kernel in the wetting zone is capable of reproducing the particle size distribution that is experimentally observed at different process conditions as well as different types of formulations, varying in hydrophilicity and API concentration. Finally, it was observed that model parameters could be linked not only to the material properties but also to the liquid to solid ratio, paving the way to create a generic PBM to predict the particle size distribution of a new formulation.

## 1. Introduction

The pharmaceutical industry started a successive shift from the traditional batch operations towards a more upgraded continuous framework. In this context, the twin-screw wet granulation has become a promising wet granulation technique in the pharmaceutical industry for continuous manufacturing of solid dosage forms due to advantages such as design flexibility, short residence time and throughput range [1]. Various works conducted on twin-screw wet granulation have investigated the aspects of the process or the equipment design that determine the quality attributes of the produced granules [2,3,4,5,6,7,8]. In contrast, less investigations have been approached to delve into the physical mechanisms that govern the process and are necessary to predict it.

As an effort to investigate the process mechanisms present in a twin-screw wet granulator, a unique dataset was collected at different locations inside the granulator and reported in [9]. It was stated that the main factor that affects the particle size distribution (PSD) is the liquid-to-solid (L/S) ratio. At low L/S ratio, the PSD exhibited bimodality, whereas at high L/S ratio the PSD showed unimodal behavior, those findings were in agreement with other works [8,10]. From that experimental data, a one-dimensional population balance model was developed for predicting the PSD at the outlet of the granulator starting from the pre-blend using aggregation and breakage as the main phenomena that take place in the granulation process [11].

Population balance modeling (PBM) has emerged as a powerful model-based tool not only to predict the output of the desired property but also to increases the understanding and the enhancement of continuous twin-screw granulation processes [12,13,14]. One-dimensional PBMs are still widely used in pharmaceutical manufacturing primarily tracking only one particle property at a time, partly due to the lack of suitable equipment capable of measuring two or more properties simultaneously, as well as the lack of suitable multi-dimensional kernels that take into account the specific process or the available experimental data [15,16].

Despite the multiple published models to simulate the twin-screw wet granulation process [1,12,13,14,15,17,18,19,20], a complete or more generic model that involves material properties, process parameters and perhaps screw configurations, is to the author’s best knowledge not available in literature [6]. This work seeks to take the first steps to address the development of a generic model, which from an industrial perspective is desirable to reduce the number of experiments required and therefore expedite the time to market for new products. In this paper, the compartmental model developed by Van Hauwermeiren et al. [11] was adapted to improve it, through an identifiability analysis. Some proposed kernels were tested, and at the end the one which presented a unique value under the threshold was chosen, resulting in an upgrade of the aggregation kernel for the wetting zone. The selected kernel was used during the initial calibration and testing process for different formulations, evidencing the need to modify the kernel once more. After a mathematical modification the final expression of the kernel was obtained. In particular, a strategy is applied to reduce the amount of unknown parameters of the model in each compartment using patterns detected from the collected experimental data.

## 2. Experimental Setup

### 2.1. Continuous Wet Granulation Experiments Using TSG

The granules analyzed in this study were produced in the high shear twin-screw wet granulator (TSWG) module of the ConsiGma^TM^-25 (GEA Pharma Systems, Collette^TM^, Wommelgem, Belgium) continuous line. The TSWG is composed of two co-rotating self-wiping screws of 25 mm diameter, with a length-to-diameter ratio of 20:1. The screw configuration comprises an initial zone of conveying elements, followed by two blocks or compartments of six kneading elements at a 60° of stagger angle each (the kneading compartments are separated by conveying elements with the same length, being 1.5 diameter of each kneading compartment), and extra conveying elements plus three size control elements right before the granulator outlet (60°). The powder pre-blend enters the granulator through a gravimetric twin-screw loss-in-weight feeder (KT20, K-Tron Soder, Niederlenz, Switzerland), and distilled water as granulation liquid is dosed into the screw chamber, using peristaltic pumps out of phase, located on top of the granulator (Watson Marlow, Cornwall, UK), connected to 1.6 mm nozzles. The addition of the liquid is done by injecting the liquid through the feeding ports, located in the upper central part of each screw. The barrel jacket temperature setpoint was 25 °C. Note that in this study the screw configuration differs from the configuration presented in the original model, as size control elements were included here to reduce large granular material and obtain a similar size among the formulations under study, which allows a fair comparison.

### 2.2. Design of Experiments

In order to study the influence of the granulation parameters such as screw speed, material throughput and liquid-to-solid (L/S) ratio, in addition to the effect of the nature of the API, on the size distribution of the resulting granules, a five-level central composite design of experiments (DoE) was performed. Three different active pharmaceutical ingredient (API) at both low and high concentrations, resulting in the study of six formulations were studied. Therefore, two or three factors were used for each formulation based on their hydrophilicity, see details in Table 1. The process conditions were chosen to operate the equipment in a stable manner as well as according to the processability of each formulation to obtain similar granules [21,22]. Formulations at low concentration contain 5% (*w*/*w*) API, 5% (*w*/*w*) hydroxypropylcellulose (The Dow Chemical Company, Midland, MI, USA), 15% (*w*/*w*), microcrystalline cellulose (Avicel^®^ PH 101, FMC, Philadelphia, PA, USA), and 75% lactose monohydrate (Lactochem^®^ Regular, DFE Pharma, Goch, Germany). Formulations at high concentration contain 50% (*w*/*w*) API, 5% (*w*/*w*) hydroxypropylcellulose (The Dow Chemical Company, Midland, MI, USA), 15% (*w*/*w*) microcrystalline cellulose (Avicel^®^ PH 101, FMC, Philadelphia, PA, USA), and 30% (*w*/*w*) lactose monohydrate (Lactochem^®^ Regular, DFE Pharma, Goch, Germany). The samples were collected at four locations inside the barrel, indicated with the red color labels in Figure 1. These measurements are possible by the usage of a second liquid addition port: this enables us to mimic the granulation behavior of these different zones at the end of the barrel so that granules can be collected continuously without the need for a screw pull-out. This method is described in full detail in the work of Verstraeten et al. [9].

It was observed that the L/S ratio is the most important factor that influences the properties of the granules for all the formulations. This is in line with what was observed by Verstraeten et al. [9] for hydrophobic and hydrophilic formulations, both of which were also studied in the present work. Higher L/S ratio conditions result in a larger number of large granules and thus there is a reduction in the amount of fine material and smaller granules. At low L/S ratios, a bimodal PSD prevails after the wetting zone. The powder is initially wetted through immersion [8] by injecting the binder liquid onto the compartment 1, called the wetting zone, where solely conveying elements are present, therefore a low–shear environment is generated. Thus part of the initial blend remains un-granulated partly due to the inability of distributing homogeneously the liquid over all the granules and on the other hand the amount of water into the system (L/S ratio condition) and the availability of water in the granules as a result of the different mechanism involved during the granule formation and the nature of the blend. Subsequently, the wetted material is redistributed, compacted [23], and deformed mostly, elongated by the kneading elements present in the kneading zones (C3 and C5 in Figure 1) due to the high-shear environment. Finally, the granules undergo extra shear and compression due to the size control elements, reducing the oversized granules.

### 2.3. Particle Size Distribution Measurements

The samples collected were dried for twenty four hours and hereafter measured using a QICPIC particle size analyzer, equipped with WINDOX 5.4.1.0 software (Sympatec, GmbH, Clausthal-Zellerfeld, Germany). Each powder PSD (dry preblend) was measured with laser diffraction using a Mastersizer^®^ 2000 (Malvern Instruments, Malvern, UK). Those measurements gave the materials initial PSD for the simulations.

## 3. Population Balance Model

### 3.1. Base Model Description

The original model is a one-dimensional PBM with granule size as internal coordinate, assuming pure aggregation in the wetting zone, other possible mechanisms were lumped into the aggregation process to avoid parameter identifiability issues in the calibration process taking into account the data available. That assumption was based on the observations from the collected data, where the PSD often exhibited a bimodal shape after the wetting zone. Those experiments performed at low L/S ratio exhibited more bimodality than those at high L/S ratio. However, since only a low-shear environment is present in the wetting zone, breakage is not likely to be the main mechanism leading to the bimodality, findings supported by the analysis of the liquid distribution performed in the experimental work by Verstraeten et al. [9]. On the other hand, a combination of aggregation and breakage was used to model the dynamics in the kneading zones. The general expression for tracking the temporal change of particle number density in a spatially homogeneous system undergoing aggregation and breakage within the gist of the framework is described in Equation (Equation 1) [24].
(1)$$\begin{array}{cc} \frac{\partial n}{\partial t} & =\frac{1}{2}\int_{0}^{x}\beta \left(x-\epsilon ,\epsilon ,t\right)n\left(x-\epsilon ,t\right)n\left(\epsilon ,t\right)d\epsilon -\int_{0}^{\infty }\beta \left(x,\epsilon ,t\right)n\left(\epsilon ,t\right)d\epsilon \\ \; & \;+\int_{0}^{\infty }b\left(x,\epsilon \right)S\left(\epsilon \right)n\left(\epsilon ,t\right)d\epsilon -S\left(x\right)n\left(x,t\right) \end{array}$$

In Equation (Equation 1), n(x,t) is the particle number density as a function of time (t) and its internal coordinate, here granule size (x) with size referring to the particle volume. The first two terms on the right-hand side of Equation (Equation 1) correspond respectively to the birth and death of particles due to aggregation dynamics: the birth of particles of size *x*, as a result of aggregation of two particles with sizes ϵ and x−ε, and the death of the original particles that were aggregated. In these first two terms, β represents the aggregation kernel. This function describes the aggregation rate of two particles ϵ and x−ε. The aggregation kernel β(t,x,ϵ) can be esteemed as a product of two factors: the aggregation efficiency $\beta_{0}\left(t\right)$ and the collision frequency $\beta \left(x,\epsilon \right):\beta \left(t,x,\varepsilon \right)=\beta_{0}\left(t\right)\beta \left(x,\varepsilon \right)$, the latter describing the transport of granules leading to collisions, the former how successful these collisions are. The last two terms denote the birth and death of particles due to breakage. In these terms, *b* represents the probability density function for the formation of particles of size *x* from a particle of size ϵ, *S* represents the selection function, which determines the rate at which particles of size *x* are selected to break.

Due to the impossibility of traditional aggregation kernels within a PBM framework, e.g., the sum (β(x,ε)=x+ε) or product kernel (β(x,ε)=x·ε) to capture the bimodal shape curve from an initially unimodal distribution a new aggregation kernel with a step-like behavior as a function of granule size was proposed in the original model by [11]. The original collision frequency is presented in Equation (Equation 2) [11].
(2)$$\begin{array}{cc} \beta \left(x,\varepsilon \right)= & \left(\frac{top_{1}}{2}\left(1+\tanh\left(\frac{R_{1}^{3}-{\left(x^{2}+\varepsilon^{2}\right)}^{{}^{1}\;{/}_{2}}}{\delta_{1}}\right)\right)\right.\\ \; & \left.-\frac{top_{1}-top_{2}}{2}\left(1+\tanh\left(\frac{R_{2}^{3}-{\left(x^{2}+\varepsilon^{2}\right)}^{{}^{1}\;{/}_{2}}}{\delta_{2}}\right)\right)\right)\\ \; & \left(x^{{}^{1}\;{/}_{3}}\varepsilon^{{}^{1}\;{/}_{3}}\right). \end{array}$$

Equation (Equation 2) is composed of two parts, a two-dimensional stepping function which is the responsible to induce bimodality, and a product kernel $\beta \left(x,\varepsilon \right)=\left(x^{1/3}\;\cdot \;\varepsilon^{1/3}\right)$ responsible of a monotonously increasing contribution. The two-dimensional stepping function (described by the first and second line of Equation (Equation 2)), in turn, is decomposed in a distance function $\left(R_{\theta }^{3}-{\left(x^{2}+\varepsilon^{2}\right)}^{1/2}\right)$, and a smoother $\left(\delta_{\theta }\right)$ as argument of the hyperbolic tangent function.

To simplify the aggregation kernel $\beta_{0}\left(t\right)$ was assumed to remain constant with respect to time and to be size independent. $R_{1}$, $\delta_{1}$, $\delta_{2}$, and $top_{2}$ were chosen to be fixed, the parameters $R_{2}$, $top_{2}$, and $\beta_{0}$ were calibrated in the compartment corresponding to the wetting zone.

In the kneading zones a linear selection function of the form $S\left(m\right)=S_{0}m^{1/3}$, where $S_{0}$ is a positive breakage rate constant and *m* represents the size of the particle before breakage. And the probability for the formation of granules of size *d* (which stands for daughter particle) after the breakage of a granule of size *m* (mother particle) represented by the breakage function b(m,d) were used. The breakage function takes into account attrition and binary breakage, represented in Equation (Equation 3) [11], this equation states that when a particle breaks, some smaller fragments are formed, the remaining part of the granule breaks into two pieces.
(3)$$b\left(m,d\right)=f_{\mathrm{prim}}\;\frac{1}{\sqrt{2\pi }\sigma }e^{-\frac{{\left(d^{\frac{1}{3}}-\mu \right)}^{2}}{2\sigma^{2}}}\frac{m}{\mu^{3}}\frac{1}{3d^{\frac{2}{3}}}+\left(1-f_{\mathrm{prim}}\right)\;\frac{2}{m}$$

In Equation (Equation 3), $f_{prim}$ represents the fraction of granules selected to form small fragments. σ is the standard deviation, and μ the mean of the Gaussian normal distribution corresponding to the particles that result from erosion. *m* and *d* are the volume of the mother and daughter particle, respectively.

### 3.2. Calibration Procedure

To calibrate the model, the Particle Swarm Optimization (PSO) method was implemented as it is a global optimizer which has been proved to be more robust with regards to local minima [25]. Calibration was performed individually for each compartment, for each experiment and for each formulation. The PSD resulting from the simulations was obtained from the best combination of each set of parameters with the minimum value of the objective function Equation (Equation 4) [26] which in turn reached the best fit with the experimental data.
(4)$$D\left(u,v\right)={\left(2E|X-Y|-E\left|X-X^{\prime }\right|-E\left|Y-Y^{\prime }\right|\right)}^{1/2}$$

The Equation (Equation 4) represents the distance between the experimental data and the model prediction *u* and *v* respectively in the notation, where *X* and $X^{\prime }$ are independent random variables whose probability distribution is *u*. Similarly, *Y* and $Y^{\prime }$ are independent random variables whose probability distribution is *v*. This equation is called the energy distance in its implementation in the Python package Scipy [27] and is a way to write down the maximum mean discrepancy. The simulations were run in different round of calibrations. The first round of calibration included three parameters in the wetting zone $\beta_{0}$, *R*, and Step and a total of five parameters in the kneading zones from the Equation (Equation 3): $S_{0}$, μ, σ and $f_{prim}$, and the aggregation efficiency $\beta_{0}$ due to a combination of breakage and aggregation was assumed. Subsequent rounds of calibrations included two parameters from the new aggregation kernel in the wetting zone, proposed in Section 4 to calibrate: the aggregation efficiency $\beta_{0}$, and Step. A total of three parameters were calibrated in the kneading zones, $\beta_{0}$, $S_{0}$, and $f_{prim}$. The remaining unknown parameters were set as suggested in Section 5.2.

It is important to highlight that the main focus of this work is presented in the development and improvement on the original model by presenting a new aggregation kernel for the wetting zone and reducing the number of unknown model parameters. Therefore, the details of the calibration results will be presented as an extension of this work in a future publication.

## 4. Identifiability

### 4.1. Introduction to Identifiability

In the work of [28], Chapter 7 on loss functions for PBMs, it was observed that some parameters of the model described in the previous section have flat regions in the loss function that could not be attributed to the behavior of the loss function itself. This is an issue for interpretability: if a simulated PSD can be attained by a range of parameter values, what is then the physical meaning of this parameter? In an ideal case, we want to have each possible model outcome to be linked with one unique parameter set. Let us introduce the concept of structural identifiability:
A model is structurally identifiable if it is possible to determine the values of its parameters from observations of its outputs and knowledge of its dynamic Equations [29].

This definition puts up two constraints: we need to be able to observe our outputs and we need to know the system equations. For the sake of simplicity, assume for the remainder of this work that we have this information. In light of this work, we have measurements of PSDs and we know the system equations, i.e., a PBM.

With some mathematical rigor, the problem can be approached in a local or a global sense. A parameter $\theta_{i}$ is structurally locally identifiable, if for almost any $\theta_{i}$, there exists a neighborhood Ω such that [30]:(5)$$\left.\begin{array}{c} {\left(\theta_{i}\right)}_{1},{\left(\theta_{i}\right)}_{2}\in \Omega \\ f\;\left({\left(\theta_{i}\right)}_{1}\right)=f\;\left({\left(\theta_{i}\right)}_{2}\right) \end{array}\right\}\Rightarrow {\left(\theta_{i}\right)}_{1}={\left(\theta_{i}\right)}_{2},$$
with *f* the system equations. Interpret *almost any*
$\theta_{i}$ as all values except isolated points. If Ω spans the entire parameter space, then $\theta_{i}$ is called structurally globally identifiable. In mathematics, this is equivalent to stating that the map from $\theta_{i}$ to $f(\theta_{i})$ is injective: $\theta_{i}\mapsto f\left(\theta_{i}\right)$. The structural identifiability of model parameters for a given set of system equations assumes error-free measurements. This is interesting for applications where approximately continuous and noise-free observations can be attained.

Alas, our world is noisy and cannot be sampled continuously, therefore an extension is needed to consider measurement noise. Furthermore, it should be taken into account that although we assume that the model framework is known, some parts of the model may only be approximately accurate in describing the system dynamics. If we add these two remarks to the definition of identifiability, we call the result practical identifiability.

The goal of this subsection is to bring the concept of practical identifiability to the world of PBMs. To the authors’ best knowledge, this is the first time that identifiability analysis is performed on PBMs.

In this work, a workflow is presented to deal with identifiability when the model outcome is a probability mass function (pmf). Further, it should be noted that identifiability is a local property on several levels. The obvious first one is local in the parameter space. Next, also local in the sense of model definition: a different kernel may have completely different behavior. In the PBM-framework, the grid needs to be defined. This grid will for sure have a profound effect on the results of an identifiability study. But the effect of the grid is out of scope for this work, and it is the same throughout this work. Last, there is a lot to say about the measurement error: different pre-processing steps, different preblends, the operator effect, different TSWG, different measurement device or settings, etc. The upside is that the data used in this entire work was collected using the same granulator, measurement device and carried out by the same person. For the sake of simplicity, it will be assumed that the measurement error is approximately constant for all experiments.

This leads us to the two aspects we are going to investigate: the parameters and the kernels. The question that will be attempted to answer is: given the parameters, kernel definition and measurement error, for which experiments can we perform a reliable parameter estimation?

### 4.2. Identifiability Approach

A new approach for performing identifiability analysis with PBMs is needed. Let us sum up what is desired in this approach: feasible calculation time, flexible in different inputs and outputs, ability to incorporate measurement error, and intuitive interpretability of the results. It would be feasible to have an approach that can determine whether a proposed kernel makes sense viewed from a parameter estimation angle, both for the kernel structure as for the choice of parameters. Next, using the measurement errors, we want to determine for which experiments the parameters can be estimated so that the confidence intervals do not overlap. In other words, which experimental conditions can be estimated independently.

In the remainder of this section, two approaches will be discussed. First, the potential issue with the aggregation kernel will be tackled by a combination of parameter space sampling and analysis of the inner workings of the population balance equations. Second, a definition of expected measurement error for our data will be given. Using that definition, new proposals for kernels definitions will be tested and an approach to determine uncertainty regions in the parameter space will be constructed.

### 4.3. Potential Kernel Issues

#### 4.3.1. The Origin of the Problem

To illustrate the staircase behavior, scenario 9 of the aforementioned research on objective functions for PBMs is used [28], Chapter 7. Note that the other scenarios described in Van Hauwermeiren [28], Chapter 7 have no relevance for the topic discussed in this paper, so they will not be mentioned here. In the so-called scenario 9, the behavior of the distance functions for a PBM model using the aggregation kernel described in Equation (Equation 2) in function of one parameter $R_{2}$ was investigated, while keeping the other parameters constant. In other words, the effect of $R_{2}$ on the model output was investigated. In Figure 2a, the normalized Maximun Mean Discrepancy (MMD) is plotted in function of said parameter $R_{2}$. The horizontal flat parts can be observed in that figure. This is not an ideal situation: this means that for some piece-wise parts of the parameter space, the distance to the optimum remains the same, i.e., the pmf does not change. But why does this happen? This might be a property of the distance function itself. But, as can be seen in Figure 2b, similar behavior is occurring in all tested distance functions (see [28], Chapter 7 for the complete overview of all distances). In that figure, the normalized function values in function of the parameter $R_{2}$ are shown. The vertical gray lines point to the beginning or the end of a step in the distance function value. Darker shades of gray indicate that several distance functions have a step located at that specific value of $R_{2}$. This figure shows that the behavior is not a property of a certain distance function, as all functions under study have similar behavior.

So, what is the origin of this behavior? To get to the answer, the aggregation kernel will be shown in more detail and how these function values are *seen* by the modelling framework of choice, PBM.

Let us consider the aggregation kernel of Equation (Equation 2) with parameters equal to the true distribution of scenario 9 in [28], Chapter 7. The values for the parameters are: $R_{2}=500$, $top_{1}=10$, and $\beta_{0}=7\times 10^{-12}$. A 2D contour plot of the aggregation kernel with these specific parameter values is shown in Figure 3: the aggregation speed is plotted in function of the size of the two aggregating particles. In that figure, a black line is drawn, that line is an indication of the domain of the 1D plots in Figure 4a,b. In these figures, the size of one of the aggregating particles is kept constant while the size of the other is varied. Figure 4a shows the aggregation kernel values without discretization of the grid, i.e., a grid with an infinitesimally small cell width. Figure 4b shows these values with discretization of the grid that is used in the PBM simulations. The parameter $R_{2}$ is varied between 400 μm and 600 μm.

What can be seen in Figure 4a is a smooth step superimposed on a function that increases with increasing particle size. This smooth step was intended as such when writing down the mathematical description of the newly formed aggregation kernel. It was postulated in [11] that this smoothness is needed for numerical stability of the ODE integrator. When looking at the discretized version of the aggregation kernel function, i.e., on the PBM grid, this smooth step does not occur, not even approximately. For one value of $R_{2}$, there exist at most two levels between the low and high point of the step. This means that for all $R_{2}$ values which fall between two representative sizes, the resulting aggregation kernel matrix does not change drastically. Interpret the aggregation kernel matrix as an N×N matrix with all combinations of two particles of the grid with length *N*, for which the aggregation kernel value is computed. So the origin of the staircase-like behavior lies in the fact that we need to discretize the grid to calculate the PBM. As stated before, the grid is fixed throughout this whole work. Increasing the number of nodes in the grid would increase the resolution to better capture the step, but this is out of the question for this work. If we cannot change the grid, how can we then cope we the issue?

Figure 5a,b give a visual cue to what particle sizes new particles can aggregate. From Figure 5a, it can be seen, that there are a lot of gray lines right of the line for a representative particle size. These lines are the particles that form as a combination of a particle with that representative size and a small particle. This can also be seen in Figure 5b as the two *legs* in the heatmap have two very similarly colored lines.

As it seems, the aggregation kernel has features whose effect cannot be seen due to the inherent need to discretize the grid. Further, in in [11], the claim was made that smooth behavior was needed for numerical stability. However, upon closer inspection of the problem, these numerical instabilities could not be reproduced. So, the smoothness is not needed for the computation of the model and its effect can hardly be seen.

#### 4.3.2. New Aggregation Kernels

Let us propose two new kernel formulations without the unwanted behavior. For completeness: the effect of the second step going to zero at 7000 μm is invisible as the largest representative size is below that value and particles cannot aggregate out of the system. The first suggestion approximates the original kernel: using a circle in the (x,ε)-plane to determine where the step is located:(6)$$\beta \left(x,\varepsilon \right)=\left\{\begin{array}{cc} x^{{}^{1}\;{/}_{3}}\varepsilon^{{}^{1}\;{/}_{3}}, & \;x^{2}+\varepsilon^{2}\leq R^{2}\\ step\;x^{{}^{1}\;{/}_{3}}\varepsilon^{{}^{1}\;{/}_{3}}, & \;x^{2}+\varepsilon^{2}>R^{2}. \end{array}\right.$$

The parameter *R* is the critical particle size: particle combinations larger than this size will aggregate faster, step is the increase in aggregation speed for particle combinations exceeding the critical size. The second alternative is instead of using a circle as a distance from the origin, the location of the step is defined as a square in the (x,ε)-plane:(7)$$\beta \left(x,\varepsilon \right)=\left\{\begin{array}{cc} x^{{}^{1}\;{/}_{3}}\varepsilon^{{}^{1}\;{/}_{3}}, & \;x\leq R\wedge \varepsilon \leq R\\ step\;x^{{}^{1}\;{/}_{3}}\varepsilon^{{}^{1}\;{/}_{3}}, & \;x>R\vee \varepsilon >R. \end{array}\right.$$

To compare the newly proposed kernels with the original kernel, the absolute relative difference between the kernels is computed. A heatmap with the results is shown in Figure 6. In that figure, it can be seen that only for three particle combinations, the error is significant.

#### 4.3.3. Analysis of the New Aggregation Kernels

If these two new kernels are applied to the same problem (scenario 9 from [28], Chapter 7) its performance can be analyzed. In Figure 7, the results using the MMD as a distance function are used. From this figure, it can be seen that the kernel in Equation (Equation 6) behaves almost identical to the original kernel. The kernel defined by Equation (Equation 7) has fewer steps and has a true non-smooth step-like behavior.

Next to the plot of the MMD in function of the parameter, it is important to look at the effect of a different kernel on the resulting PSD. Because only then it can be assessed whether the significant difference of three particle combinations (shown in Figure 6) has a significant effect on the PSD. Consider a region bounded by the MMD values: from the global minimum to the minimum value of the kernel with the square step. The PSDs corresponding with step-values within that region are plotted in Figure 8a,b. In these figures, it can be seen that the PSDs do not differ much. With some expert knowledge, it can be said that all these distributions are within measurement error. This would mean that the kernel definition that leads to unambiguous results should be chosen, i.e., the kernel with the square step.

### 4.4. Incorporating Measurement Error into Parameter Estimations

Before the bold statement “*this looks to be within measurement error*” can be made, it requires a more rigorous approach. It is difficult to come up with measurement errors on the type of data we are using. On the other hand, in [28], Chapter 7, it was shown that there exist good definitions of the distance between two data points, i.e., two pmfs, for the set up under study. From repeated PSD measurements and measurements of repeated experiments, pairwise MMD distances are calculated. The median MMD of this collection of values gives a good indication of what experiments can be distinguished with the current measurement technology. The median is chosen over the mean to be more robust to outliers in the data. A visualization of these repeated measurements is given in Figure 9.

In Figure 10, this indication of measurement error is plotted on top the MMD in function of the parameter *R*, the same plot that was shown in Figure 7. In this figure, it can be seen that the only kernel with a single, unique value below the measurement error is the kernel defined in Equation (Equation 7).

Thus, it would make sense to simplify the kernel in the wetting zone from Equation (Equation 2) to Equation (Equation 7) for several reasons:The original explanation of needing the smoothed step for numerical reasons does not hold, ergo it is not needed anymore.The new kernel has only two parameters (*R* and step), whereas the original kernel has six, which means a significant simplification for a parameter estimation problem.The new kernel has a single unique value below the measurement error threshold: this excludes ambiguity for choosing the right value.

As a final remark, the kernel is a piece-wise function, so to avoid some computational hurdles, the parameter *R* should be chosen out of some discrete levels instead of changed continuously. In this setup, the parameter *R* can be chosen out of the representative values of the bins. Moreover, due to its clear interpretation, it can just be directly estimated from the data instead of calibrated which further simplifies the parameter estimation problem that will be tackled in the next section.

## 5. Results and Discussion

This section presents the results of the implementation of the new aggregation kernel resulting after the identifiability analysis, as well as an extra modification that was necessary to implement it for all formulations. In addition, a strategy to reduce the number of unknown model parameters and its results is shown.

### 5.1. New Aggregation Kernel for the Wetting Zone

As it was referred in the previous Section 4, the original aggregation kernel for the wetting zone was improved by an identifiability analysis. Using parameter space mapping and measurement error thresholds, a new kernel was selected which has a smaller number of parameters while at the same time still exhibiting the desired behavior. These decisions were made based on a thorough understanding of the modeling framework and incorporating information on measurement error.

A selection of calibrated PSDs in the wetting zone using the new kernel are visualized in Figure 11 and Figure 12 corresponding to a hydrophobic formulation (API1) and a hydrophilic formulation (API2) respectively. For all scenarios, the bimodal distribution is dominant. At the lowest L/S ratio, the first peak of smaller sizes is more noticeable. However, from these figures, it is possible to elucidate a limitation in the resulting PSD for both types of formulations for both concentrations at the different liquid to solid ratio conditions. The distance between both peaks was not possible to achieve correctly at the same time, even though the height of each peak was captured correctly most of the time. Two possible causes arose here, first related to the parameter *R* included in the kernel (Equation (Equation 7)). Due to its physical meaning, which represents the critical particle size, where particles larger than this size will aggregate significantly faster than those that are smaller, and represents where the step is located or in other words, the beginning of the transition from small particles to big agglomerates. The value of *R* thus determines the mode of the first peak. Low *R* values mean that the first peak will be at lower particles sizes, moreover, it is related to the features of the initial blend (nature of the blend) that will be discussed in the next Section 5.2. The second, is related to the value of the power in the expression.

From the preliminary results of the calibration of the parameters *R* and Step, as well as $\beta_{0}$ in the wetting zone, the resulting PSD that were obtained are shown in Figure 13. As a result of low values of *R*, that is, smaller particles or non-granulated material located further to the left, the second peak in the distribution, which means that the fraction corresponding to the largest granules during the simulation, exhibited a shift to the left for both hydrophobic and hydrophilic formulation at low and high concentration of the API. The initial API 1 50% pre-blend was found to exhibit a wider distribution located in higher sizes, as can be seen in Figure 11 a and b, the distance between both peaks of the resulting distribution for that specific formulation is smaller with respect to the other formulations and, being achieved for some of the experiments. The power of the expression in the kernel (Equation (Equation 7)) was included as a parameter to be calibrated once the *R*-value was set as explained in Section 5.2, thus modifying the expression resulting in a new aggregation kernel Equation (Equation 8), with the same capabilities to obtain a bimodal response from an initial unimodal distribution, granting it greater flexibility to achieve at the same time the location of both peaks for a different nature of the formulation and process conditions, aiming towards the development of a generic model capable of predicting the final distribution of granules from the properties of the material.
(8)$$\beta \left(x,\varepsilon \right)=\left\{\begin{array}{cc} \left(x^{2/5}\;\cdot \;\varepsilon^{2/5}\right), & \;x\wedge \varepsilon <R\\ \;\mathrm{step}\;\;\cdot \;\left(x^{2/5}\;\cdot \;\varepsilon^{2/5}\right), & \;x\vee \varepsilon >R \end{array}\right.$$

### 5.2. Reducing Unknown Model Parameters

A comparison of the PSD measurements in the wetting zone under different L/S ratio conditions for each formulation is presented in Figure 14. A strong trend of a decreasing fraction of the first peak was detected as the L/S ratio increases for API 1 5 and 50% as well as for API3 at 50% concentration, while a weak trend was found in API2. Furthermore, the granule size distribution progresses from left to right when the L/S ratio increases in the second peak, neglecting the noise of some experiments. However, regardless of the trend that each formulation follows or the condition of the L/S ratio, a pattern was identified at the location of the first peak for all formulations. The depletion point towards larger sizes is always placed in the same size category, ergo, allowing to determine the *R*-value recalling the definition given above, which is represented by the vertical black line for each formulation in Figure 14 and their values are shown in Table 2. The *R*-value for the API1 5% and API2 5% formulations were found identical and this fact is associated with the similarity in the formulation due to the low API content and the composition of fixed excipients. Since the hydrophilicity increases from API1 to API3, which means that API3 is the most hydrophilic in the set of formulations studied, it was observed that the *R*-value decreased with hydrophilicity comparing the formulations with the same API concentration. Note that this entails that, according to this work, the *R*-value is only affected by the nature and concentration of the API. Process parameters such as L/S ratio, throughput, screw speed or barrel fill level have no significant effect. Taking the values of the parameter *R* as input for the simulations, the only parameters that remain to be calibrated in the wetting zone are Step and $\beta_{0}$. The simulated PSD in the wetting zone can be seen in Figure 15, Figure 16 and Figure 17.

The general performance of the new aggregation kernel in the wetting zone is satisfactory, for all the formulations the location of both peaks was achieved and the experimental behavior was captured. As the L/S ratio increased, the percentage of fine material decreased due to the availability of the binder liquid between the particles, therefore, a larger portion of PSD in small size categories is expected to be aggregated in large particles [8], leading to a first reduced peak in the distribution of that compartment, and even increasing the presence of fractions of intermediate sizes between both peaks. This effect was also captured by the model and was more pronounced for the API3. In most cases, nonetheless, the simulated PSDs presented a dip between both parts of the distribution, which can be interpreted as an underprediction of the intermediate size fractions. On the other hand, the results proved that the strategy adopted to reduce the number of unknown parameters satisfied all the formulations studied, in addition, the complexity of the calibration procedure was reduced. Furthermore, the observed trend of the parameter that was set can be taken as an opportunity to link the model parameter with the features of the blend. Regarding the kneading zones (C3, C5 and C6), as concluded in the original model, a clear trend was not observed for the parameters μ and σ during calibration, in the present work similar results were obtained [11]. Therefore, a strategy similar to that applied in the wetting zone was implemented after comparing the experimental data collected at different process conditions. A comparison at different L/S ratios for formulations with 50% of API concentration in each compartment is visualized in the Figure 18, where the black vertical line symbolizes the μ parameter defined in Section 3 in Equation (Equation 3) as the mean of the Gaussian normal distribution corresponding to the particles that result from erosion, namely, the mean size of the smallest particles within the resulting distribution after the breakage that occurs in the kneading zones due to the high-shear environment. Note that the position of the mean value of the remaining fine material and the smaller particles, corresponds to the same value attained for the *R* parameter from the aggregation kernel in the wetting zone. Then, μ was consequently set for all the compartments using the values shown in Table 2. Assuming neither high nor low dispersion σ was set equal to 50, leaving as a result only three parameters to be calibrated in the kneading zones: $\beta_{0}$, $S_{0}$, and $f_{prim}$. The formulations with a concentration of 5% suggested a pattern akin to those of high concentration. In order to evaluate the use of these assumptions, calibrations were performed, representative results can be seen in Figure 19. For experiments at low L/S ratio the resulting distribution exhibited a bimodal behavior less stark at C6, contrarily to the experiments at high L/S ratio conditions, where a uni-modal distribution was obtained even from C3. For all scenarios, the model could accurately capture the location of the peaks. Furthermore, the model is able to capture the shifts in the resulting distribution in each compartment due to the applied shear and compression forces inside the kneading block and the redistribution of liquid binder within the powder [8,31,32].

Overall, the achieved results using fixed parameters are satisfactory, as for formulations it was possible to reproduce the experimental behavior, capturing the final positions of the distributions and reduce the complexity of the calibration process, which in turn reduces identifiability issues. This work therefore, constitutes a leap forward in the development of a generic model for the twin-screw granulator. In the next steps of the work, some improvements are to be made: The patterns identified to establish the model parameters could be further investigated to understand their relationship with the process conditions or material properties and build sub-models that,when integrated, allow the construction of a more generic and and even more robust model.

## 6. Conclusions

In this work, a new aggregation kernel was presented to obtain the granule size distribution in the wetting zone for a twin-screw wet granulation process based on the model by Van Hauwermeiren et al. [11]. A complementary strategy based on the principle of identifiability was used to reduce the unknown parameters of the model not only in the wetting zone but also in the kneading zones. The results proved that the new kernel combined with the reduction of unknown parameters can simulate the observed experimental behavior, capturing accurately the position of both peaks in the wetting zone as well as simulating the effects of screw configuration on the distributions in the kneading zones. This model is a necessary improvement aimed at the development of a generic tool to track the size distribution of granules along the barrel as well as at the outlet of the twin-screw granulator. 

## Figures and Tables

**Figure 1 pharmaceutics-13-00692-f001:**
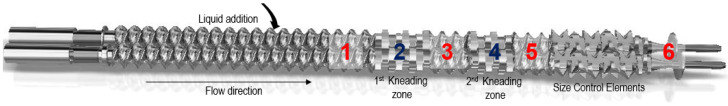
Twin-screw configuration.

**Figure 2 pharmaceutics-13-00692-f002:**
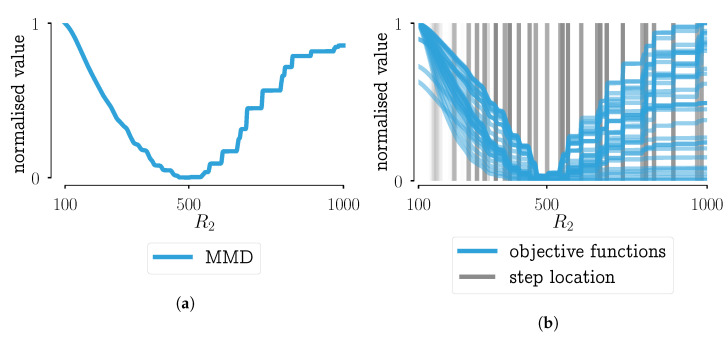
Plotting of the objective function values in function of the parameter *R*_2_ for scenario 9 (PBM simulation with 35 bins), as shown in [28], Chapter 7 Results for objective functions using (**a**) the MMD, (**b**) all objective functions that were under study. The vertical lines indicate where a step in the objective function value is located. Darker shades of grey indicate that more objective functions have a step on that location. The values shown in the figures are normalized to allow straightforward comparison between objective functions.

**Figure 3 pharmaceutics-13-00692-f003:**
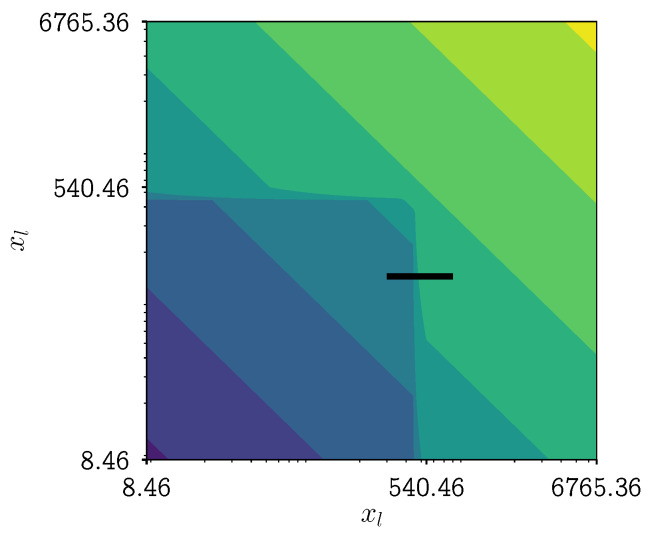
Contour plot of the aggregation kernel values of scenario 9 in [28], Chapter 7 in function of the size of the two particles that aggregate. The black line indicates the domain of the 1D plot in Figure 4a,b.

**Figure 4 pharmaceutics-13-00692-f004:**
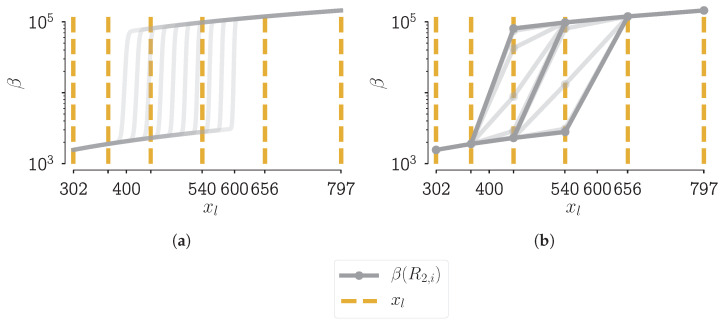
1D plot of the aggregation kernel values of scenario 9 in [28], Chapter 7 in function of the size *x*_l_ . The size of one of the aggregating particles is kept constant while the size of the other particle is varied. An indication of the domain of this 1D plot can be seen as black line in Figure 3. The gray lines in these figures denote the aggregation kernel function values in function of size for a fixed value of *R*_2_. The parameter *R*_2_ is varied between 400 and 600. For clarity, the representative sizes are indicated with yellow dashed lines. Figure (**a**) shows the aggregation kernel values without discretization of the grid, i.e., a grid with an infinitesimal small cell width. Figure (**b**) shows these values with discretization of the grid that is used in the PBM simulations.

**Figure 5 pharmaceutics-13-00692-f005:**
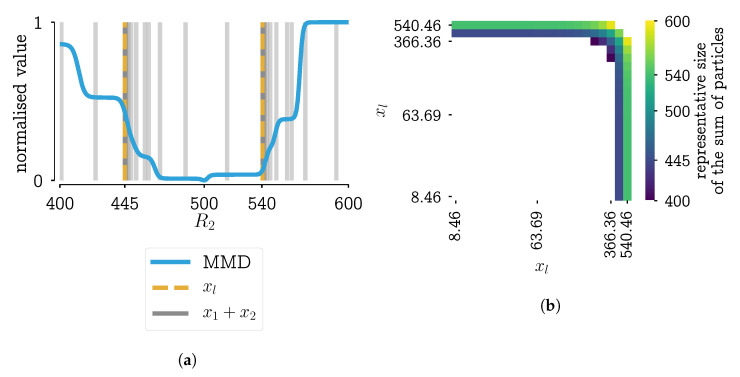
Figure (**a**) shows a part of Figure 4a with added vertical grey lines illustrating the different sizes of new particle which can be formed out of two smaller particles by aggregation. The aggregates lie between 400 μm and 600 μm. Figure (**b**) shows where these aggregates can be found in a similar fashion as the 2D contour plot in Figure 3. The aggregation smaller than 400 μm and larger than 600 μm are masked out of this figure.

**Figure 6 pharmaceutics-13-00692-f006:**
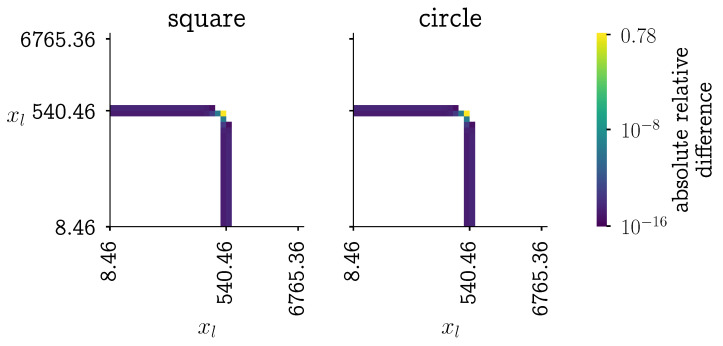
Heatmap of the relative difference between the aggregation kernel defined Equation (Equation 2) and the newly proposed kernels in Equations (Equation 6) and (Equation 7). The relative difference equal to zero are masked out. The location of the step is at 500 μm.

**Figure 7 pharmaceutics-13-00692-f007:**
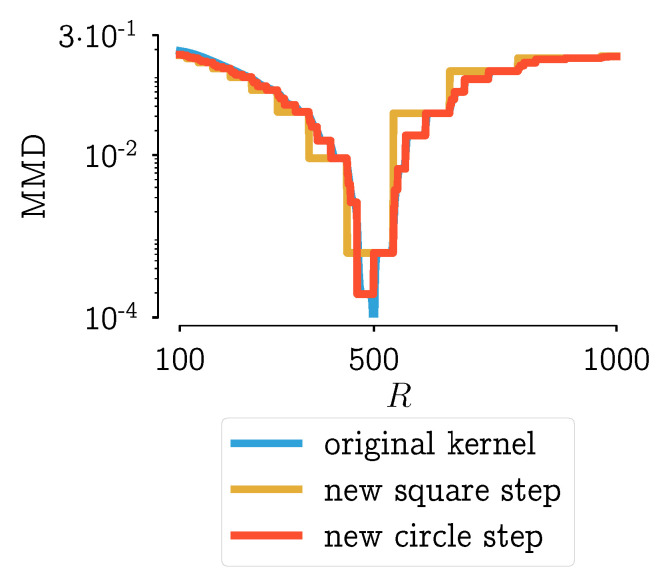
Plotting of the objective function values in function of the parameter step for scenario 9 (PBM simulation with 35 bins), as shown in [28], Chapter 7. Results for objective functions using the MMD (same as Figure 2a) and the new kernels which use a circle or square as a distance function.

**Figure 8 pharmaceutics-13-00692-f008:**
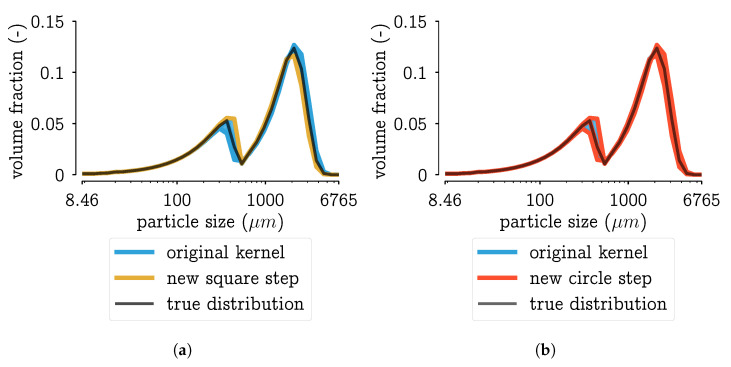
Plotting the PSDs corresponding to the parameter range bounded by the MMD values in Figure 7: from the global minimum to the minimum value of the kernel with the square step. Figure (**a**) shows the aggregation kernel with the square step defined by Equation (7), Figure (**b**) the kernel with the circle step defined by Equation (7). In each figure, the PSDs of the original kernel are plotted as well.

**Figure 9 pharmaceutics-13-00692-f009:**
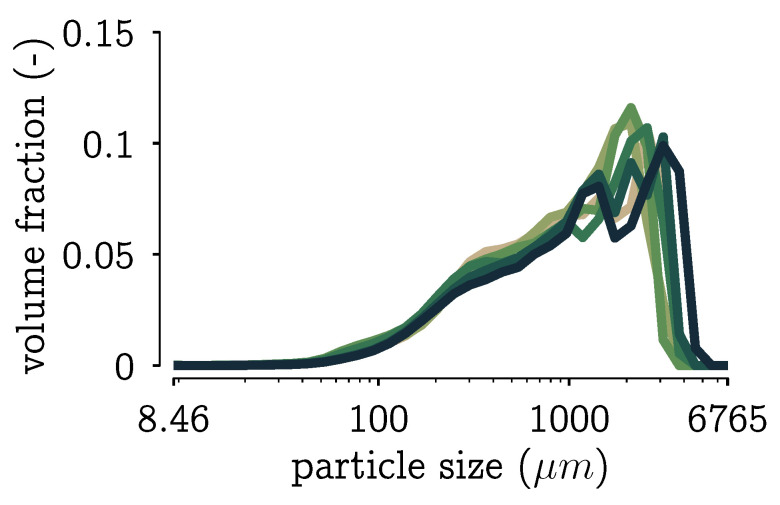
Visualization of measured PSDs in volume fraction for some repeated experiments and repeated measurements of the same experiment. Note that the measurements get noisier for larger particles.

**Figure 10 pharmaceutics-13-00692-f010:**
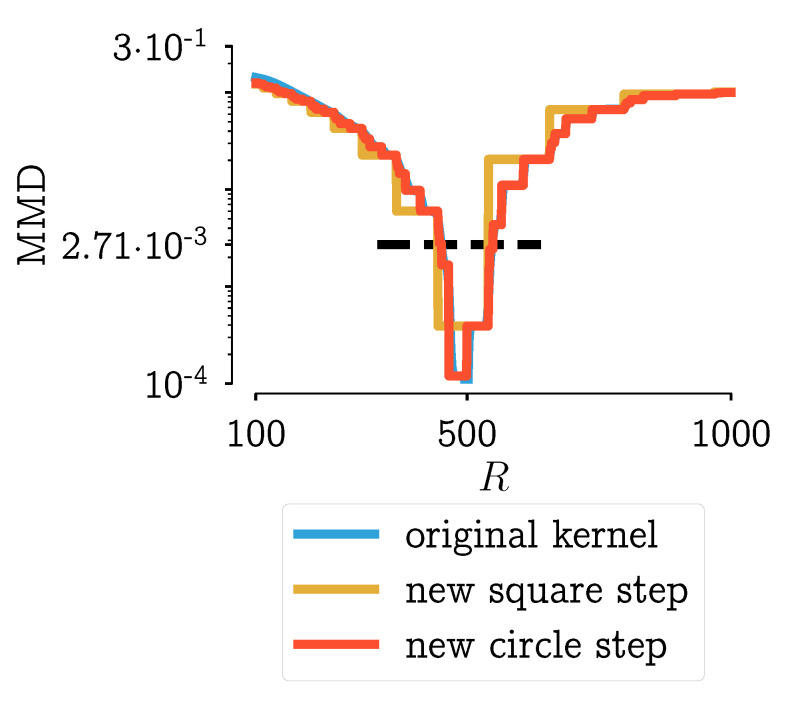
Plotting of the objective function values in function of the parameter step for scenario 9 (PBM simulation with 35 bins), as shown in [28], Chapter 7 Results for objective functions using the MMD and the new kernels which use a circle or square as a distance function. The horizontal black line is an indication of the measurement error: the average distance between repeated measurements or measurements of repeated experiments.

**Figure 11 pharmaceutics-13-00692-f011:**
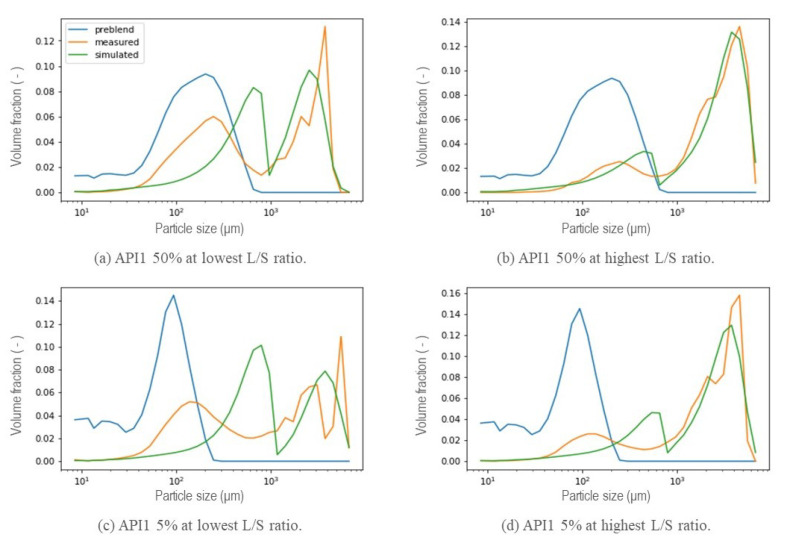
Resulting PSD in the wetting zone using the new aggregation kernel at different L/S ratio conditions for a hydrophobic formulation (API1).

**Figure 12 pharmaceutics-13-00692-f012:**
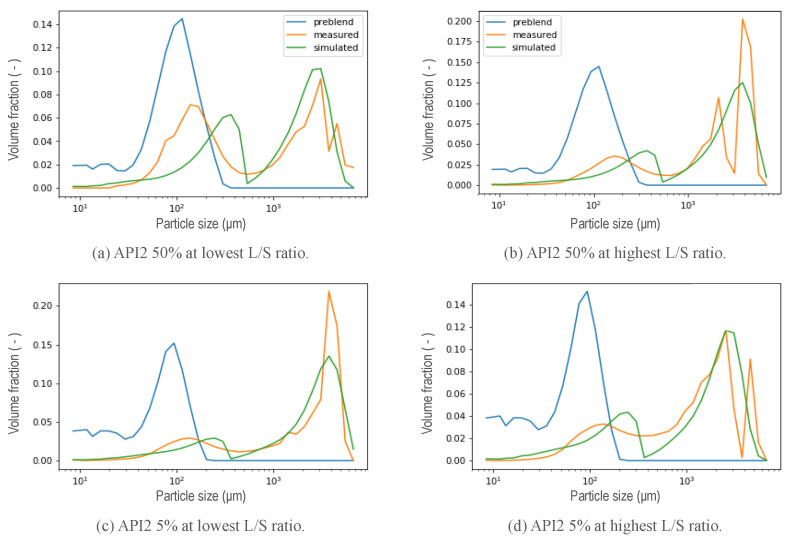
Resulting PSD in the wetting zone using the new aggregation kernel at different L/S ratio conditions for a hydrophilic formulation (API2).

**Figure 13 pharmaceutics-13-00692-f013:**
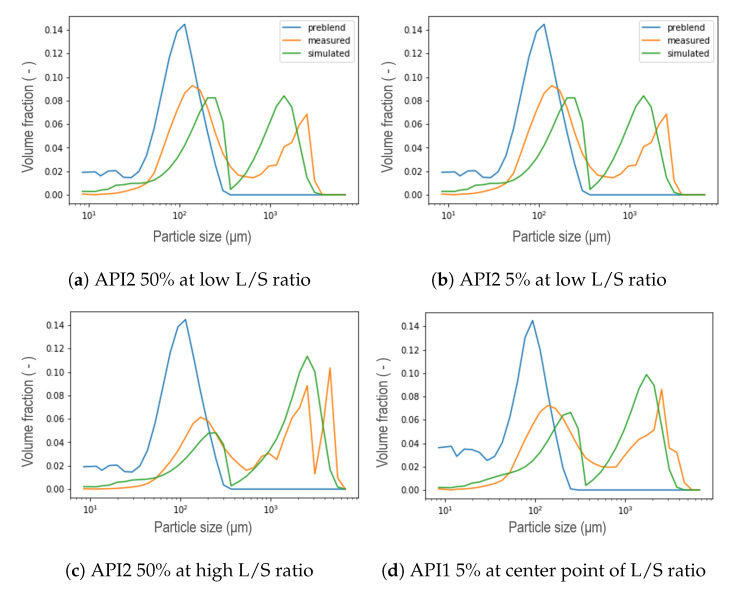
Preliminary results of the calibration using the new kernel at different process conditions of the DoE for API1 and API2 at low and high concentrations in the wetting zone.

**Figure 14 pharmaceutics-13-00692-f014:**
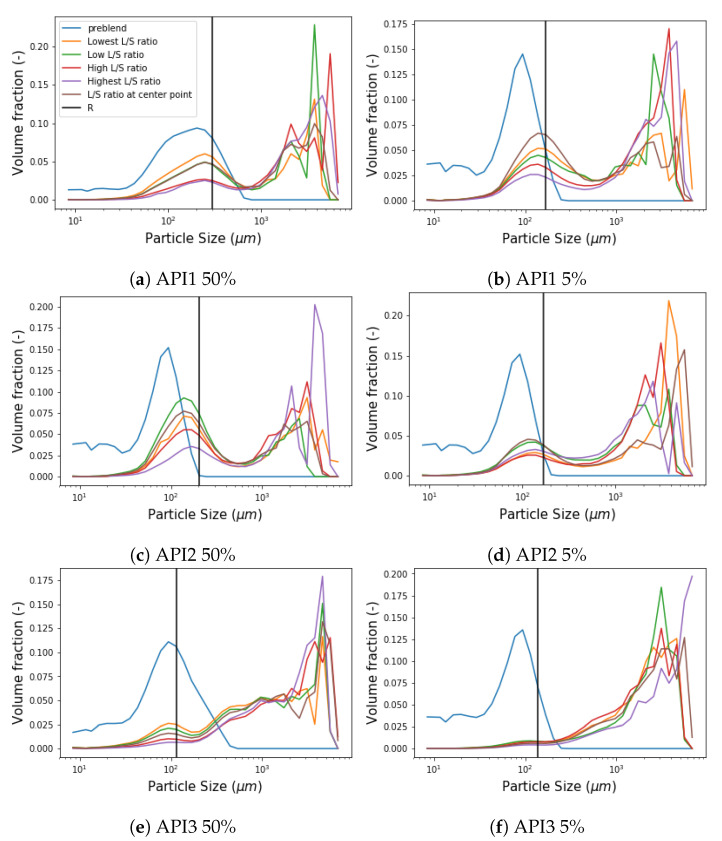
Comparison among different L/S ratios of the resulting PSD in the wetting zone for each formulation showing its respective *R*-value.

**Figure 15 pharmaceutics-13-00692-f015:**
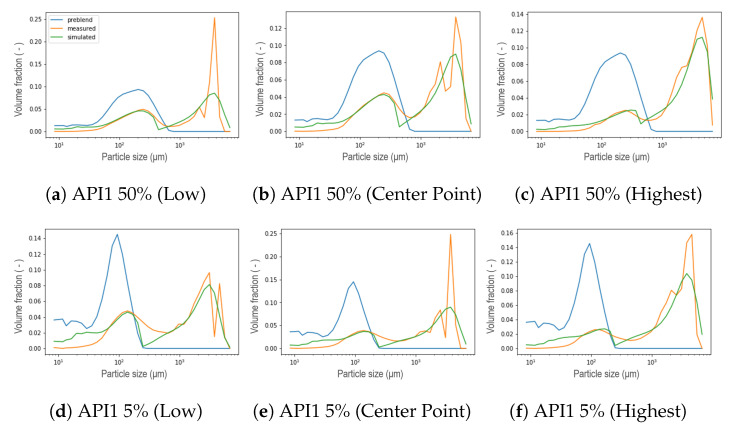
Resulting PSD in the wetting zone for API1 at different L/S ratio conditions within the DoE.

**Figure 16 pharmaceutics-13-00692-f016:**
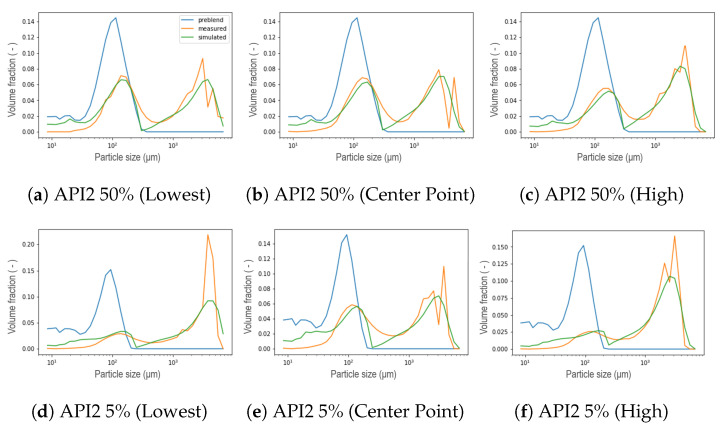
Resulting PSD in the wetting zone for API2 at different L/S ratio conditions within the DoE.

**Figure 17 pharmaceutics-13-00692-f017:**
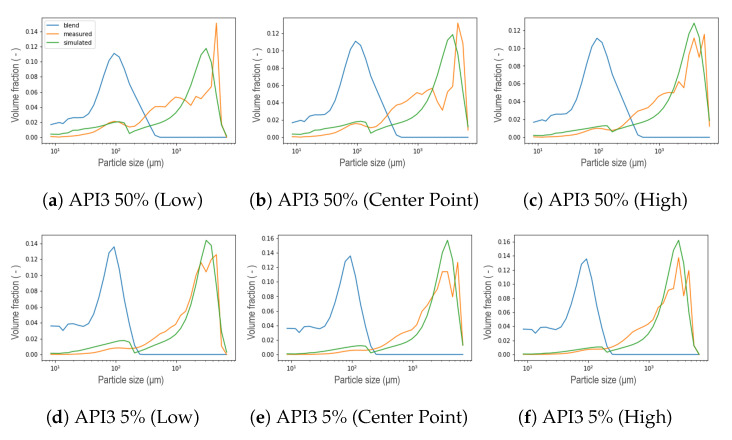
Resulting PSD in the wetting zone for API3 at different L/S ratio conditions within the DoE.

**Figure 18 pharmaceutics-13-00692-f018:**
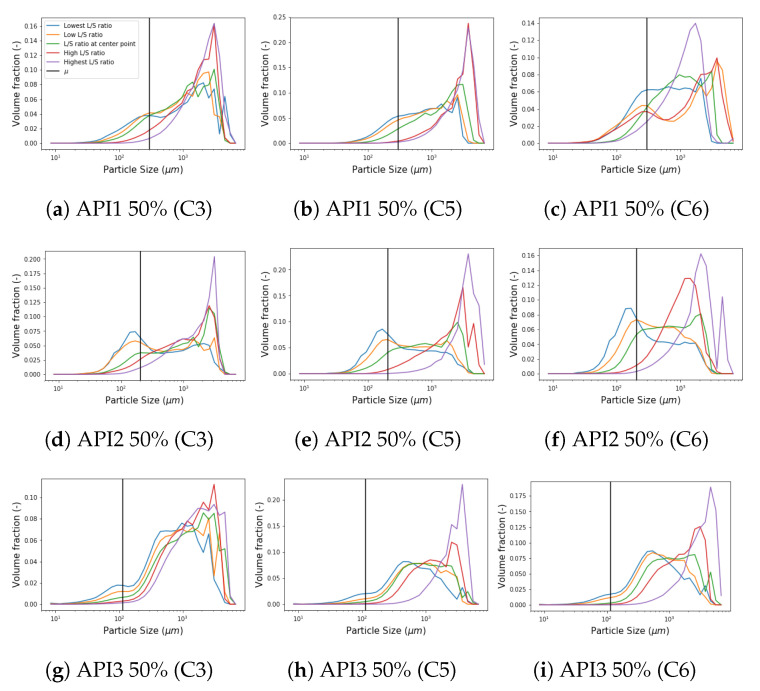
Comparison of the experimental PSD for formulations with high concentration at different L/S ratio conditions within the DoE in each compartment.

**Figure 19 pharmaceutics-13-00692-f019:**
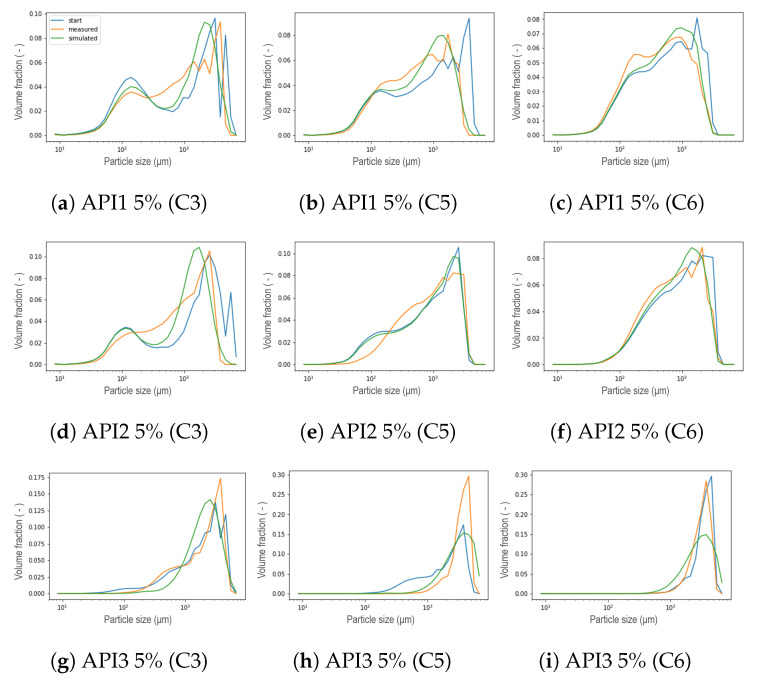
Representative results of the calibration taking μ and σ constant for each formulation. (**a**–**c**) were simulated at low L/S ratio for API1 5%. (**d**–**f**) were obtained with the L/S ratio at the center point within the DoE for API2 5%. (**g**–**i**) were simulated at high L/S ratio condition for API3 5%.

**Table 1 pharmaceutics-13-00692-t001:** Process conditions of the experimental work (DoE).

API	Number of Experiments	Screw Speed (rpm)	Throughput (kg/h)	L/S (%)
API1 (5%)	11	675	15–25	8.9–20.2
API1 (50%)	11	675	15–25	13.6–23.7
API2 (5%)	17	450–675	15–25	8–18
API2 (50%)	17	450–675	15–25	5.2–16
API3 (5%)	17	750–900	15–25	15.2–18.5
API3 (50%)	17	450–675	15–25	5.2–13.4

**Table 2 pharmaceutics-13-00692-t002:** *R*-values for each formulation.

Formulation	R(m)
API1 (5%)	168.35
API1 (50%)	301.64
API2 (5%)	168.35
API2 (50%)	204.47
API3 (5%)	138.61
API3 (50%)	114.12

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
