# Peer review of "Improvement of a 1D Population Balance Model for Twin-Screw Wet Granulation by Using Identifiability Analysis"

_pharmaceutics, 2021, doi:10.3390/pharmaceutics13050692_

Round 1
Reviewer 1 Report
Generally, the present manuscript is an interesting one that investigated the continuous TSWG, which is a key unit operation in the OSD continuous manufacturing. A 1D population balance model was improved by a new aggregation kernel modeling approach. The study increased process understanding of the continuous manufacturing. However, the background introduction definitely needs to be improved by including more recent and relevant publications. For example, the Yoon group’s (Umass Lowell, U.S. http://sites.uml.edu/syoon/publications/ ) work should be included and given credit. They have done a lot of experimental studies on the continuous TSWG, which has covered all the process parameters (through put, screw speed and L/S ratio) investigated in the study. They even developed a compartmental PBM which is very similar to the model in the present manuscript. This should be included and discussed in the introduction. Minor comments are as follows:
- Section 2.3: As I know, the QicPic is able to get both PSD and mean size, so how is the Mastersizer® 2000 used in the present study?
Author Response
Response to Reviewer 1 Comments
We would like to thank the reviewers for their useful feedback. It allowed us to further improve the quality of the manuscript. Please find detailed answers to the questions raised in this rebuttal letter.
Generally, the present manuscript is an interesting one that investigated the continuous TSWG, which is a key unit operation in the OSD continuous manufacturing. A 1D population balance model was improved by a new aggregation kernel modeling approach. The study increased process understanding of the continuous manufacturing.
However, the background introduction definitely needs to be improved by including more recent and relevant publications. For example, the Yoon group’s (Umass Lowell, U.S. http://sites.uml.edu/syoon/publications/ ) work should be included and given credit. They have done a lot of experimental studies on the continuous TSWG, which has covered all the process parameters (through put, screw speed and L/S ratio) investigated in the study. They even developed a compartmental PBM which is very similar to the model in the present manuscript (it was included, reference 19). This should be included and discussed in the introduction. Minor comments are as follows:
- Section 2.3: As I know, the QicPic is able to get both PSD and mean size, so how is the Mastersizer® 2000 used in the present study?
Response 1: The Mastersizer 2000 was used to obtain the PSD of the preblend. It has a broader measuring range (i.e., able to measure smaller particles) and higher resolution. It gives a better dispersion of raw materials compared to the Qicpic that was used to obtain the granule PSD. Furthermore, an identical method was used by Verstraeten et al. (2017), on which the current experimental study builds further.

Reviewer 2 Report
Summary: The authors have presented a study to improve the aggregation kernel in the wetting zone and reduce the number of parameters for the kernels for wetting and mixing zones. The paper is well written and the writing style is particularly appealing and palpable. The authors have made a novel contribution to the literature. Some details are needed to improve the reading of the manuscript as listed below:
- Why was a significantly higher screw speed used for API3 5% formulation?
- It seems that the L/S ratio range is quite different for different formulations. What was the range of L/S ratio for each formulation based on?
- I suggest the authors also cite the following papers that look at granulation mechanisms along the barrel for kneading element configurations:
- Li et al., Chemical Engineering Science 113 (2014) 11–21
- Li et al., Powder Technology 346 (2019) 363–372
- Li et al., International Journal of Pharmaceutics 496 (2015) 3–11
- I suggest the authors cite the following references related to size control elements and granule size limits in twin screw wet granulation
- Portier et al., International Journal of Pharmaceutics 576 (2020) 1189812
- Pradhan et al., Powder Technology 315 (2017) 290–299
- Lines 252-254 refer to a scenario 9 whereas Figures 3 and 4 indicate scenario 8. This difference is appearing multiple times in the manuscript. Please fix.
- In the materials and methods section, please include a description of how the granule samples were collected from compartments C1, C3, and C5. Similarly, the differences in hydrophilic/hydrophobic nature can be stated upfront in the experimental section.
- Since there are several references to Scenario 9 in the PhD thesis. I suggest including a short section in the manuscript summarizing scenario 9.
- It is not clear to me if the experimental PSD in figures 11-13 are from the wetting zone (compartment 1 in Figure 1. Please specify under the figures.
- From Figure 13, as the R value was reduced the fines peak of the simulation successfully moved to the left, but it looks like the coarse peak also moved to the left. Can the authors discuss the implication of R on the coarse mode size?
- In Figure 14, the axes scale and fonts are hard to read. I suggest, please increase the size of the font.
- In Figure 19, what does the “start” PSD refer to?
- The current screw configuration utilizes the 60 Forward angle of the kneading elements. However, backmixing is usually observed in the wetting zone when reverse kneading elements are used resulting in potentially less bimodal distribution and more liquid dispersion. Can the authors comment on how these effects can be incorporated/ what changes need to be made in the strategy discussed in this paper?
- R is also likely dependent on fill level in the granulator. Can the authors suggest how R can be figured when the granulator fill level changes?
Author Response
Response to Reviewer 2 Comments
We would like to thank the reviewers for their useful feedback. It allowed us to further improve the quality of the manuscript. Please find detailed answers to the questions raised in this rebuttal letter.
Reviewer 2
Summary: The authors have presented a study to improve the aggregation kernel in the wetting zone and reduce the number of parameters for the kernels for wetting and mixing zones. The paper is well written and the writing style is particularly appealing and palpable. The authors have made a novel contribution to the literature. Some details are needed to improve the reading of the manuscript as listed below:
- Why was a significantly higher screw speed used for API3 5% formulation?
Response 1: From preliminary trials with that formulation it was seen that some conditions produced torque overload, therefore, the screw speed, as well as the other process conditions, were selected to avoid torque overload during the execution of the DoE.
Added to the document in the section 2.2: The process conditions were chosen to operate the equipment in a stable manner as well as according to the processability of each formulation to obtain similar granules.
- It seems that the L/S ratio range is quite different for different formulations. What was the range of L/S ratio for each formulation based on?
Response 2: As was referred in the previous numeral, the process conditions were obtained based on the processability of each formulation. Since the focus of the work presented was focused on the perspective of simulation and modeling, it is expected that the experimental findings will be presented in another publication.
- I suggest the authors also cite the following papers that look at granulation mechanisms along the barrel for kneading element configurations:
- Li et al., Chemical Engineering Science 113 (2014) 11–21 : Understanding wet granulation the kneading block of twin-screw extruders.
Incorporated in the experimental setup section.
- Li et al., Powder Technology 346 (2019) 363–372: Granule transformation in a twin-screw granulator: Effects of conveying, kneading, and distributive mixing elements.
Incorporated in the results and discussion section.
- Li et al., International Journal of Pharmaceutics 496 (2015) 3–11: Examining drug hydrophobicity in continuous wet granulation within a twin-screw extruder.
Incorporated in the experimental setup section.
- I suggest the authors cite the following references related to size control elements and granule size limits in twin screw wet granulation
- Portier et al., International Journal of Pharmaceutics 576 (2020) 1189812: Continuous twin screw granulation: A complex interplay between formulation properties, process settings, and screw design
Incorporated in the experimental setup section.
- Pradhan et al., Powder Technology 315 (2017) 290–299:
Other work by this Author has already been considered and included.
- Lines 252-254 refer to a scenario 9 whereas Figures 3 and 4 indicate scenario 8. This difference is appearing multiple times in the manuscript. Please fix.
Response 5: The typos were adjusted throughout the document.
- In the materials and methods section, please include a description of how the granule samples were collected from compartments C1, C3, and C5. Similarly, the differences in hydrophilic/hydrophobic nature can be stated upfront in the experimental section.
Response 6: These measurements at compartments C1, C3, C5 are possible by the usage of a second liquid addition port and combining this with specific placement or absence of kneading zones: this enables us to mimic the granulation behavior of these different zones at the end of the barrel so that granules can be collected continuously without the need for a screw pull-out. This method is based on of Verstraeten et al. (2017). We have added this additional information in the paper (second paragraph of section 2.2).
- Since there are several references to Scenario 9 in the PhD thesis. I suggest including a short section in the manuscript summarizing scenario 9.
Response 7: We have made a selection of information and visuals from these chapters of the PhD thesis so that the identifiability can be explained in the most concise way. The authors feel that the train of thought expressed by going from an explanation on the identifiability (section 4.1 and 4.2), to the problem and solution (section 4.3 and 4.4) stands on its own. Using the information in section 3.1 on the population balance equation (equation 1) and the kernel under study (equation 2), which parameters are altered and which are kept constant (section 4.3.3), we feel that the problem is described in full detail without any ambiguity.
- It is not clear to me if the experimental PSD in figures 11-13 are from the wetting zone (compartment 1 in Figure 1. Please specify under the figures.
Response 8: Suggestion implemented. Indeed, those graphics correspond to the results in the wetting zone.
- From Figure 13, as the R value was reduced the fines peak of the simulation successfully moved to the left, but it looks like the coarse peak also moved to the left. Can the authors discuss the implication of R on the coarse mode size?
Response 9: From the analysis of the data, It was found that R is exclusively related to the position of the fine or smaller material, it means it just determines the position of the first peak in the case of the bimodal distributions. From compartment 1 to 3 on, R becomes then, in that case, the latest parameter refers to the position of the first peak as well, and due to the kneading elements, breakage is present. Therefore the distribution undergoes a reshape, which in turn shifts to the left.
- In Figure 14, the axes scale and fonts are hard to read. I suggest, please increase the size of the font.
Response 10: Some attempts were made, however, due to the final composition of the figure, the individual graphs lost readability. Therefore, probably you can no see a big difference. Individual plots are available.
- In Figure 19, what does the “start” PSD refer to?
Response 11: The "start" distribution represents the initial particle size distribution in the corresponding compartment. In other words, it is the final distribution measured in the immediately preceding compartment that becomes the initial distribution for the next compartment simulation.
- The current screw configuration utilizes the 60 Forward angle of the kneading elements. However, backmixing is usually observed in the wetting zone when reverse kneading elements are used resulting in potentially less bimodal distribution and more liquid dispersion. Can the authors comment on how these effects can be incorporated/ what changes need to be made in the strategy discussed in this paper?
Response 12: The current PBM is only calibrated for a specific screw configuration. In this case, a limited amount of additional experiments on a selection of formulations is necessary to evaluate the structure of the current model and identify whether it is still valid or needs to be adapted. However, as demonstrated, the model in the wetting zone with a single aggregation kernel can capture the bimodality variations, with a low or high Step parameter. Additionally, all these possible phenomena would be lumped in the parameter calibration instead of explicitly modeling them. And that could be fine as long as the predictive power remains high.
- R is also likely dependent on fill level in the granulator. Can the authors suggest how R can be figured when the granulator fill level changes?
Response 13: In the experimental DoE, we have changed the L/S ratio, the screw speed, and the mass flow rate. By altering the screw speed and/or the mass flow rate, the granulator fill level changes. Since we have found that the value of parameter R is only a function of the nature of the API and its concentration, the fill level has - in this particular setup - no significant effect. We have added a few lines in the first paragraph of section 5.2 to stress this result.

Reviewer 3 Report
I have carefully read the paper of Jiménez et al., which concerns the application of a 1D population balance model for twin-screw wet granulation by using identifiability analysis. In particular, this paper presented a comparison with previous PBM methodologies and offered an improvement. As a first step, the authors have proposed two new kernels for powder aggregation, analyzed them and finally showed the effect of them on the simulating PSD. The simulation was able to catch the PSD peaks but not their distance. So the author decided to create a further kernel in this way: they use the structure of Eq 7, but choosing the power based on a calibration procedure. This procedure led to a good matching between simulations and experiments. My opinion is that the paper is eligible to be published after a revision. In particular, I would suggest the authors to improve the readability of the manuscript: - In the introduction, I would introduce bullet points of the procedure followed by the authors to reach the final kernel - Section 4.3.1: the authors refer to Van Hauwermeiren 2020, which is a PhD thesis. The paper should be self-standing. If possible, the authors can provide data and further explanations as supporting information. - Section 4.3.3: same, scenario 9? - lines: 321-324: Not clear to me. Please explain. - line 332: Not clear to me why the new kernel has a single unique value below the measurement error threshold. What is the difference between the three curves in Fig.10? - If I have understood correctly, R changes with formulation and concentration w/w but not with L/S. Is it correct? - How did the authors calibrate step and beta by taking into account the difference L/S? - The authors calibrated the kernel over a single (maybe three, see question above) points over an experimental PSD. It turned that this calibration led to a good match between the PSD obtained with simulations and the experiments. But, apparently, the model was not validated. Typos: - MMD, missing acronym for the maximum mean discrepancy - The authors should follow the same citation rule in the whole paper: Figure/figure/Fig.; Section/section; Equation/equation/Eq.; - Please revise the format of Eq.6,7,8 - Section 5.1,5.2: eliminate colons from the title - lines 396,398: the format "R-value" is different from the previous ones (e.g., line 380). - line 104: space between "present," and "therefore" - line 308, 314: i.e: add commaAuthor Response
Response to Reviewer 3 Comments
We would like to thank the reviewers for their useful feedback. It allowed us to further improve the quality of the manuscript. Please find detailed answers to the questions raised in this rebuttal letter.
Reviewer 3
I have carefully read the paper of Jiménez et al., which concerns the application of a 1D population balance model for twin-screw wet granulation by using identifiability analysis. In particular, this paper presented a comparison with previous PBM methodologies and offered an improvement. As a first step, the authors have proposed two new kernels for powder aggregation, analyzed them and finally showed the effect of them on the simulating PSD. The simulation was able to catch the PSD peaks but not their distance. So the author decided to create a further kernel in this way: they use the structure of Eq 7, but choosing the power based on a calibration procedure. This procedure led to a good matching between simulations and experiments. My opinion is that the paper is eligible to be published after a revision. In particular, I would suggest the authors to improve the readability of the manuscript:
- In the introduction, I would introduce bullet points of the procedure followed by the authors to reach the final kernel
Response: Additional information was added to the final paragraph of the introduction section (lines 52 to 62 of the new numeration).
- Section 4.3.1: the authors refer to Van Hauwermeiren 2020, which is a PhD thesis. The paper should be self-standing. If possible, the authors can provide data and further explanations as supporting information.
Response: We have made a selection of information and visuals from these chapters of the PhD thesis so that the identifiability can be explained in the most concise way. The authors feel that the train of thought expressed by going from an explanation on the identifiability (section 4.1 and 4.2), to the problem and solution (section 4.3 and 4.4) stands on its own. Using the information in section 3.1 on the population balance equation (Equation 1) and the kernel under study (Equation 2), which parameters are altered and which are kept constant (section 4.3.3), We believe that the problem is described in such a way that it provides the ideas without losing the main focus of the presented document.
- Section 4.3.3: same, scenario 9? - lines: 321-324: Not clear to me. Please explain.
Response: The first paragraph of section 4.3.3 explains what `scenario 9` means. We have made a few additions to increase readability and to ensure that the explanation is self-standing, in line with the previous comment of the reader.
- line 332: Not clear to me why the new kernel has a single unique value below the measurement error threshold. What is the difference between the three curves in Fig.10?
Response: This is explained in the final paragraph of section 4.4. Because the new suggested kernels are piece-wise functions, parameter R is chosen out of some discrete levels instead of changed continuously. In this setup, the parameter R can be chosen out of the representative value of the bins. In other words, you limit the number of different values that R can take. In that way, the different “flat regions” in figure 10 are lumped into 1 choice for R value, thus eliminating the ambiguity of the parameter value. In this way we achieve our goal: a parameter estimation problem involving the estimation of parameter R has a unique solution.
- If I have understood correctly, R changes with formulation and concentration w/w but not with L/S. Is it correct?
Response: Correct, we have found that the value of parameter R is only a function of the nature of the API and its concentration. We have added a few lines in the first paragraph of section 5.2 to stress this result.
- How did the authors calibrate step and beta by taking into account the difference L/S?
Response: In the experimental DoE, we have changed the L/S ratio, the screw speed and the mass flow rate. Resulting in the number of experiments per formulation mentioned in the Table 1. The calibration process was performed for each experiment for each formulation in each compartment.
- The authors calibrated the kernel over a single (maybe three, see question above) points over an experimental PSD. It turned that this calibration led to a good match between the PSD obtained with simulations and the experiments. But, apparently, the model was not validated.
Typos:
- MMD, missing acronym for the maximum mean discrepancy
The missing acronym is included
- The authors should follow the same citation rule in the whole paper: Figure/figure/Fig.; Section/section; Equation/equation/Eq.;
Thank you for your comment. It was unified.
- Please revise the format of Eq.6,7,8
The format of these equations has been unified.
- Section 5.1,5.2: eliminate colons from the title
It was corrected. The colons were eliminated.
- lines 396,398: the format "R-value" is different from the previous ones (e.g., line 380).
It was corrected.
- line 104: space between "present," and "therefore"
It was implemented.
- line 308, 314: i.e: add comma
It was implemented.
